# Topological magnons driven by the Dzyaloshinskii-Moriya interaction in the centrosymmetric ferromagnet Mn$_5$Ge$_3$

M. dos Santos Dias [1,2,3] ✉, N. Biniskos [4,10] ✉, F. J. dos Santos [5,6] ✉, K. Schmalzl [7], J. Persson[8], F. Bourdarot [9], N. Marzari [5,6], S. Blügel [1], T. Brückel [8] & S. Lounis [1,2]

The phase of the quantum-mechanical wave function can encode a topological structure with wide-ranging physical consequences, such as anomalous transport effects and the existence of edge states robust against perturbations. While this has been exhaustively demonstrated for electrons, properties associated with the elementary quasiparticles in magnetic materials are still underexplored. Here, we show theoretically and via inelastic neutron scattering experiments that the bulk ferromagnet Mn$_5$Ge$_3$ hosts gapped topological Dirac magnons. Although inversion symmetry prohibits a net Dzyaloshinskii-Moriya interaction in the unit cell, it is locally allowed and is responsible for the gap opening in the magnon spectrum. This gap is predicted and experimentally verified to close by rotating the magnetization away from the $c$-axis with an applied magnetic field. Hence, Mn$_5$Ge$_3$ realizes a gapped Dirac magnon material in three dimensions. Its tunability by chemical doping or by thin film nanostructuring defines an exciting new platform to explore and design topological magnons. More generally, our experimental route to verify and control the topological character of the magnons is applicable to bulk centrosymmetric hexagonal materials, which calls for systematic investigation.

Recent breakthroughs in the physics of electrons in solids resulted from the application of topological concepts to the quantum-mechanical wave function, highlighting the role of the Berry phase[1]. For instance, the modern understanding of the integer quantum Hall effect in a two-dimensional (2D) system is that of a gapped bulk with non-zero Chern numbers which imply the existence of chiral edge states responsible for the quantized conduction[2]. Three-dimensional (3D) topological insulators are gapped by the spin-orbit interaction, leading to Dirac-like surface states with a linear dispersion and spin-momentum locking that underpin the quantum spin Hall effect[3]. There is now a systematic classification of the possible topological phases in electronic systems, encompassing also gapless systems such as Weyl semimetals, where the dimensionality but also spatial and magnetic symmetries are prominent[4–6].

[1]Peter Grünberg Institut and Institute for Advanced Simulation, Forschungszentrum Jülich & JARA, D-52425 Jülich, Germany. [2]Faculty of Physics, University of Duisburg-Essen and CENIDE, D-47053 Duisburg, Germany. [3]Scientific Computing Department, STFC Daresbury Laboratory, Warrington WA4 4AD, UK. [4]Forschungszentrum Jülich GmbH, Jülich Centre for Neutron Science at MLZ, Lichtenbergstr. 1, D-85748 Garching, Germany. [5]Laboratory for Materials Simulations, Paul Scherrer Institut, 5232 Villigen, PSI, Switzerland. [6]Theory and Simulation of Materials (THEOS), and National Centre for Computational Design and Discovery of Novel Materials (MARVEL), École Polytechnique Fédérale de Lausanne, 1015 Lausanne, Switzerland. [7]Forschungszentrum Jülich GmbH, Jülich Centre for Neutron Science at ILL, 71 Avenue des Martyrs, F-38000 Grenoble, France. [8]Forschungszentrum Jülich GmbH, Jülich Centre for Neutron Science (JCNS-2) and Peter Grünberg Institut (PGI-4), JARA-FIT, D-52425 Jülich, Germany. [9]Université Grenoble Alpes, CEA, IRIG, MEM, MDN, F-38000 Grenoble, France. [10]Present address: Charles University, Faculty of Mathematics and Physics, Department of Condensed Matter Physics, Ke Karlovu 5, 121 16 Praha, Czech Republic. ✉e-mail: m.dos.santos.dias@fz-juelich.de; nikolaos.biniskos@matfyz.cuni.cz; flaviano.dossantos@psi.ch

Topology is agnostic to whether the (quasi)particles are fermions or bosons, so that magnons can also be responsible for novel physical effects[7]. Perhaps the first example is the Hall effect experienced by a thermally-induced magnon current in ferromagnetic (FM) $Lu_2V_2O_7$, with the Dzyaloshinskii-Moriya interaction (DMI) playing a similar role to the one of spin-orbit coupling (SOC) for electronic systems[8–10]. Magnetic materials with an hexagonal lattice generally exhibit a Dirac-like magnon dispersion at the K-point in the Brillouin zone, which if gapped signals the existence of non-trivial topology[11–15]. Such topological magnon insulators were experimentally identified in 2D FM materials[16–18], while gapless Dirac magnons were characterized in 3D magnetic materials such as antiferromagnetic (AFM) $Cu_3TeO_6$[19] and $CoTiO_3$[20,21], and the elemental FM Gd[22]. While a symmetry-based approach has been proposed to predict materials hosting topological magnons[23], experimentally confirming the topological character is challenging, as the magnon Hall effect is difficult to measure and other signatures such as a characteristic winding of the scattering intensity have only recently been detected[21].

In this article, we study the 3D centrosymmetric ferromagnet $Mn_5Ge_3$[24,25]. This material exhibits significant anomalous Hall[26] and Nernst[27] effects which are signatures of large electronic Berry phases. Its Curie temperature ($T_C$) can be enhanced by carbon doping[28] as successfully described by a previous theoretical work[29], but its magnonic properties remain unexplored. Here, we theoretically predict and experimentally confirm the existence of gapped Dirac magnons at the K-point due to the DMI. This gap can be closed by rotating the magnetization direction with an external magnetic field, thus validating the proposed gap mechanism and confirming the topological character of the magnons at the K-point. Our experimental route to verify and control the topological character of the magnons is not limited to $Mn_5Ge_3$ and should also be applicable to other bulk centrosymmetric hexagonal materials.

## Results

### Basic properties of $Mn_5Ge_3$

A high-quality $Mn_5Ge_3$ single crystal of about 10 g has been grown using the Czochralski method. The space group is $P6_3/mcm$ and the unit cell contains 10 Mn atoms (and 6 Ge atoms), with Mn1 and Mn2 occupying the Wyckoff positions $4d$ and $6g$, respectively[24]. In the $ab$-plane, Mn1 adopts a honeycomb lattice while Mn2 adopts an hexagonal arrangement, Fig. 1a. Along the $c$-axis, Mn1 forms chains and Mn2 columns of face-sharing octahedra, Fig. 1b. The Curie temperature ($T_C$) is around 300 K and the magnetic moments of the Mn1 and Mn2

atoms are 1.96(3) $\mu_B$ and 3.23(2) $\mu_B$, respectively, aligned along the $c$-axis[24,25].

### Magnetic properties from first principles

Density functional theory (DFT) calculations were performed prior to the inelastic neutron scattering measurements in order to provide an initial picture of the expected magnon excitations and to identify interesting regions in (**Q**,$E$) space. The theoretical magnetic moments from juKKR (2.11 $\mu_B$ and 3.14 $\mu_B$ for Mn1 and Mn2, respectively) and the computed Heisenberg exchange interactions are comparable to the ones previously reported[29], as seen in Table 1. Spin-orbit coupling leads to significant DMI (c.f. Table 1), much weaker symmetric anisotropic exchange (not included in Eq. (1)), and a small uniaxial magnetic anisotropy energy ($K \approx -0.1$ meV), so that the relevant spin Hamiltonian (with $|\mathbf{S}_i| = 1$) reads:

$$\mathcal{H} = K \sum_i (S_i^z)^2 - \sum_{i,j} J_{ij} \mathbf{S}_i \cdot \mathbf{S}_j - \sum_{i,j} \mathbf{D}_{ij} \cdot (\mathbf{S}_i \times \mathbf{S}_j). \tag{1}$$

Here $J_{ij}$ are the Heisenberg exchange interactions and $\mathbf{D}_{ij}$ are the DMI vectors which mostly align along the $c$-axis, with the strongest shown in Fig. 1b. We discovered that some of the magnetic interactions, namely the AFM ones, are quite sensitive to small changes in the unit cell parameter and the atomic positions. The impact of this can be seen in the theoretical magnon dispersions shown in Fig. 1c, where we compare the results obtained with the experimental crystal structure parameters and with the theoretically optimized ones. In both cases there are two magnon bands in the energy range of experimental interest, with an energy gap at the K-point where otherwise a Dirac-like crossing of the bands would be expected by symmetry. This is straightforwardly verified to arise from the DMI, as omitting it from the magnon calculation leads to the closing of the gap.

### Dzyaloshinskii-Moriya interaction in centrosymmetric systems

The DMI is the key magnetic interaction for the subsequent interpretation of our experimental findings, so before we continue we wish to clarify how it can be present and have an effect in a centrosymmetric material. In his seminal paper[30], Moriya established the symmetry rules that the interaction named after Dzyaloshinskii and himself must obey. The most famous of these rules is that if two spins are connected by an inversion center then the respective DMI must identically vanish. This pointedly explains why we find finite DMI in our calculations for $Mn_5Ge_3$: it is finite for those spin pairs that do not contain an inversion

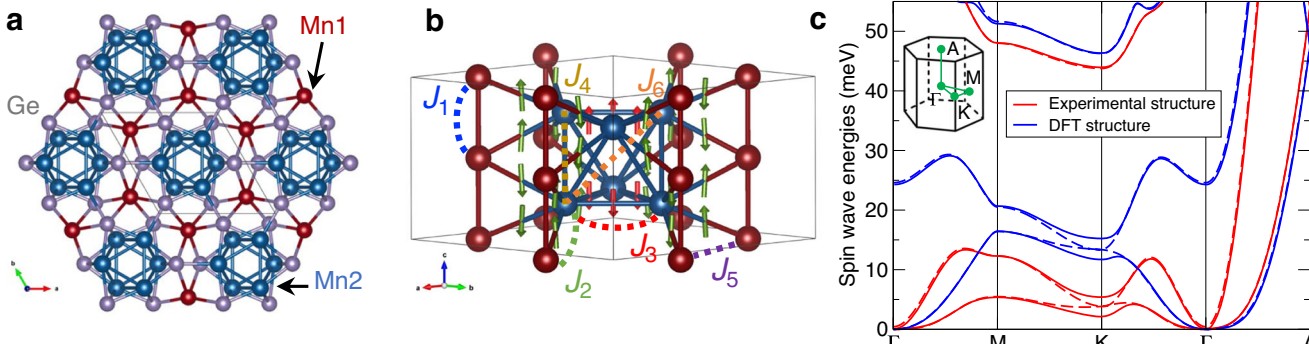

**Fig. 1 | Crystal structure and theoretical magnon bands of $Mn_5Ge_3$. a** Top view of the crystal structure, showing bonds with length up to 4.3 Å. The unit cell is indicated by the thin black lines. **b** Perspective view of the crystal structure, omitting the Ge atoms. The first few magnetic exchange interactions are marked with the respective non-vanishing Dzyaloshinskii-Moriya interaction (DMI) vectors located in the bond centers. The values are collected in Table 1. **c** Computed magnon bands using the magnetic exchange interactions extracted from DFT calculations

performed for the DFT optimized structure (blue) and for the experimental structure reported in ref. 24 (red). The solid and dashed lines show the magnon bands obtained with and without the DMI, respectively. With DMI, a gap opens at the K-point. The magnetization is set along the $c$-axis, which is the easy-axis of the material. The inset indicates the location of the high-symmetry points in the hexagonal Brillouin zone.

**Table 1 | Magnetic exchange interactions in Mn₅Ge₃ from DFT**

| | | Ref. 29 | | Experimental structure[24] | | DFT structure | |
|---|---|---|---|---|---|---|---|
| | Type | Distance (Å) | Value (meV) | Distance (Å) | Value (meV) | Distance (Å) | Value (meV) |
| $J_1$ | Mn1-Mn1 | 2.526 | 29.1 | 2.527 | 30.87 | 2.485 | 31.59 |
| $J_2$ | Mn1-Mn2 | 3.068 | 8.0 | 3.065 | 8.65 | 3.021 | 7.82 |
| $J_3$ | Mn2-Mn2 | 2.974 | −2.0 | 2.983 | −1.27 | 3.013 | −0.21 |
| $J_4$ | Mn2-Mn2 | 3.055 | 6.9 | 3.058 | 6.84 | 3.033 | 6.10 |
| $J_5$ | Mn1-Mn1 | 4.148 | −1.4 | 4.148 | −3.86 | 4.112 | −2.52 |
| $J_6$ | Mn2-Mn2 | 4.263 | 9.4 | 4.271 | 9.97 | 4.276 | 9.86 |
| $|D_2|$ | Mn1-Mn2 | – | – | 3.065 | 0.57 | 3.021 | 0.59 |
| $|D_3|$ | Mn2-Mn2 | – | – | 2.983 | 0.50 | 3.013 | 0.45 |

We compare our results using the experimental crystal structure[24] with those of ref. 29, and to our results using the theoretically optimized crystal structure. Positive (negative) values characterize FM (AFM) coupling. The corresponding pairs are indicated in Fig. 1b. The calculated magnetic interactions are long-ranged and only the first few values are given in the table. The listed values are significant to the displayed decimal precision.

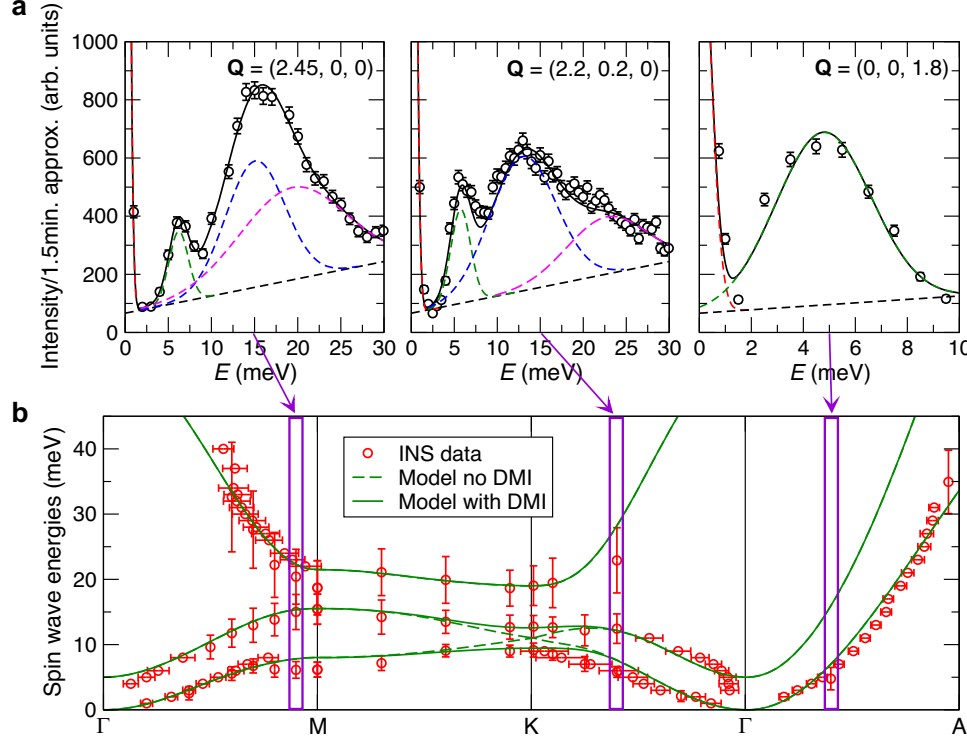

**Fig. 2 | Magnon bands of Mn₅Ge₃ determined by inelastic neutron scattering.**
**a** Representative measurements obtained at IN22 with constant-**Q** scans (symbols) at $T = 10$ K with $|\mathbf{k}_f| = 2.662$ Å⁻¹ and fits used to determine the excitation energies (lines). The purple arrows and rectangles indicate the peak positions in the corresponding (**q**,$E$) region of the dispersion. The dashed lines show the individual contributions of the various Gaussian peaks and of the linear background. The error bars indicate one standard deviation (square root of the neutron counts). **b** Experimentally determined magnon bands (symbols) and fit to a simplified model described in the text (lines). The solid and the dashed lines show the magnon bands obtained with and without the Dzyaloshinskii-Moriya interaction (DMI), respectively. The error bars indicate the uncertainty in the peak location from least-squares fitting.

center in the midpoint of the corresponding bond, such as those illustrated in Fig. 1b.

Centrosymmetry does ensure that the net DMI of the unit cell is zero, which is also verified in our simulations. This means that the ferromagnetic domain walls are not chiral and that magnetic skyrmions cannot form, in agreement with Neumann's principle. In contrast, magnons can still be influenced by the local DMI. In a semiclassical picture, the spins at different sites precess with different phases and/or amplitudes, so that certain pairs of spins are noncollinear and can be affected by the torque arising from the DMI. This is a strong effect at the K-point, where two magnon modes with opposite chirality cross and the

degeneracy is lifted in a non-perturbative way by the DMI. We now report the experimental observation of this effect and its implications.

**Magnons from inelastic neutron scattering**

Inspired by these theoretical predictions, the experimental magnon spectrum of Mn₅Ge₃ was investigated by INS. Several constant-**Q** and constant-$E$ scans have been performed at $T = 10$ K along the reciprocal space directions $(h, 0, 0)$, $(h, 0, 2)$, $(h, h, 0)$, and $(0, 0, l)$, with representative examples shown in Fig. 2a. The measured scattering intensity (circles) was fitted with Gaussian line shapes on top of a sloping background (lines).

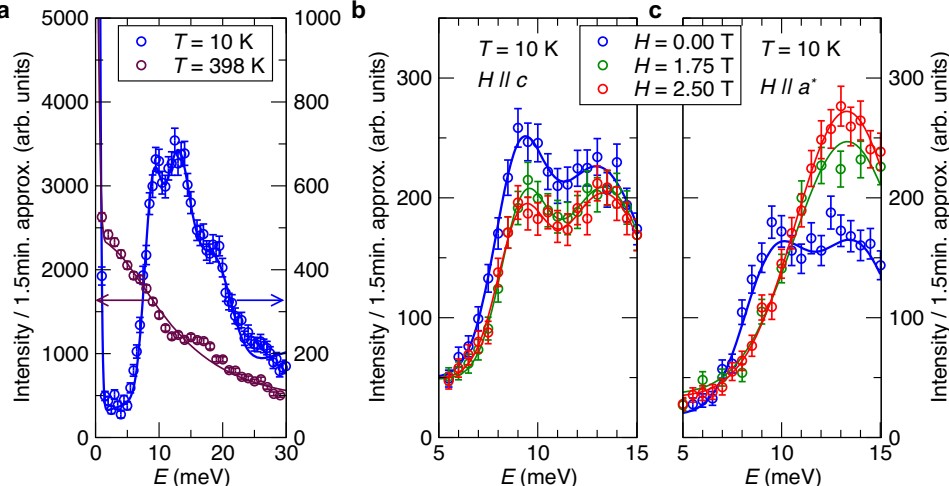

**Fig. 3 | Temperature and magnetic field dependence of the magnon peaks at the K-point determined by inelastic neutron scattering. a** Temperature dependence. At $T = 10$ K three peaks are seen between 8 meV and 22 meV. At $T = 398$ K (above $T_C$) only a broad quasielastic signal is observed. Neutron intensity for the data at 10 K and 398 K is given on the right and left axis, respectively. **b** Dependence on magnetic field applied along the $c$-axis at $T = 10$ K. The field has almost no effect on the location of the two peaks. **c** Dependence on magnetic field applied along the $a^*$-axis at $T = 10$ K. In zero field the magnetization is along the $c$-axis and two peaks are visible. The application of a transverse magnetic field saturates the magnetization along the $a^*$-axis and merges the two peaks into one, demonstrating the closure of the magnon energy gap. The data shown in **a** were obtained at IN22 ($|\mathbf{k}_f| = 2.662$ Å$^{-1}$) and the data in **b–c** at IN12 ($|\mathbf{k}_f| = 1.971$ Å$^{-1}$). The error bars indicate one standard deviation (square root of the neutron counts).

The magnetic nature of the excitations has been confirmed through their temperature dependence (see Supplementary Figs. 4–8 and Fig. 3a below), and the obtained **q** and $E$ position is given as red symbols in Fig. 2b. For the in-plane directions ($\Gamma - M - K - \Gamma$) one can distinguish the presence of three modes: two acoustic-like modes dispersing upwards in energy away from $\Gamma$ and additionally a third higher-energy mode with a steep dispersion along $\Gamma - M$ and weakly dispersive along $K - M$. In contrast, in the same investigated $E$ range a single stiff spin-wave mode is observed for the out-of-plane direction ($\Gamma - A$).

We now turn to the theoretical interpretation of these measurements. Comparing the experimental results of Fig. 2b with Fig. 1c, we find that the spin model derived from the DFT calculations qualitatively reproduces several features. The absence of a clearly visible gap in the spin-wave spectrum near zero energy transfer ($\Gamma$-point) agrees with the expected weakness of the uniaxial magnetic anisotropy[25], as also computed from DFT. The theoretical dispersion along $\Gamma - A$ is slightly stiffer than the experimental one, and a simulation of the INS intensity reveals that the second mode which is higher in energy should be invisible (see Supplementary Fig. 14). We find rather poor quantitative agreement between theory and experiment in the $\Gamma - K - M$ plane, which is probably related to the already-identified strong dependence of the magnetic interactions computed from DFT on small changes of the structural parameters. We verified that this dependence is systematic by considering various deformations of the unit cell (see Supplementary Figs. 11 and 12). However, the most crucial feature is observed both in theory and in experiment, that is the existence of a gap at the K-point where two magnon bands should otherwise cross.

In order to provide a more quantitative description of the experimentally obtained magnon bands for the in-plane directions, we constructed a simplified effective spin model (additional details are given in the Supplementary Information, Section II.E). We replace each Mn2 triangle at a constant height in the unit cell with a single effective spin $S = 9/2$ ($S = 1$ for Mn1), so that the column of Mn2 octahedra is replaced by a spin chain. Seen from the $c$-axis, the unit cell for this model thus contains two Mn1 chains and one effective Mn2 chain, and we determine the model parameters using the measured magnon energies at the high-symmetry points. The lines in Fig. 2b show that the results of this model approach indeed provide a realistic band dispersion, and confirm once more that the gap at the K-point is a consequence of a finite DMI. The model results also highlight a peculiarity of the measured magnon energies in the vicinty of $\Gamma$, to which we shall briefly return in the Discussion.

## Closing the topological magnon gap

Although there are strong theoretical arguments in favor of the topological character of the magnon gap at the K-point, a convincing experimental demonstration is in order. The gap is expected to arise from the DMI, but such a microscopic interaction cannot easy be manipulated experimentally. However, it is known that the impact of the DMI on the FM magnon spectrum depends on the relative alignment between the vectors that characterize this interaction and the FM magnetization[31–33]. Adapting these arguments to the hexagonal symmetry of Mn$_5$Ge$_3$ leads to the prediction that the magnitude of the gap should depend on the relative alignment of the FM magnetization and the crystalline $c$-axis, and in particular should vanish if the two are perpendicular. We have verified that the magnon gap closes both in the DFT-based spin model and in the one fitted to the experimental measurements when the magnetisation is perpendicular to the $c$-axis (as it does when disregarding the DMI). The magnon dispersions computed with DMI and setting the magnetization along the $a^*$-axis are identical to the dashed lines shown in Fig. 1c and Fig. 2b, which were obtained by excluding the DMI from the calculations.

This hypothesis can be experimentally tested by applying an external magnetic field. First we rule out the possibility of a phonon contribution to the inelastic peaks at the K-point, by heating the sample above its $T_C$, as seen in Fig. 3a, and verifying that the peaks disappear conforming their magnetic origin. The measurements reported so far in this work were performed in zero field, for which the magnetization is parallel to the $c$-axis due to the uniaxial magnetic anisotropy. Applying a magnetic field along the $c$-axis should lead to a very small Zeeman shift of the magnon energies, and this is indeed what we observed, as seen in Fig. 3b. If a magnetic field of similar magnitude is applied along the $a^*$-axis it overcomes the magnetic anisotropy energy and saturates the magnetization[25]. The results

obtained in this way are shown in Fig. 3c. Now the effect of the field cannot be explained by a simple Zeeman shift, and instead we find the anticipated closing of the magnon gap. The two peaks observed in zero field coalesce into a single one with an integrated intensity approximately matching the sum of intensities for the two peaks in zero field (see also Supplementary Fig. 10), although shifted to slightly higher energy than anticipated. The distinct response of the magnon excitation to a magnetic field applied to orthogonal crystal directions is consistent with the DMI mechanism, and so the gapped Dirac magnons at the K-point should consequently have a topological nature.

### Ruling out alternative explanations for a gap at the K-point
Next we rule out potential alternative mechanisms to the DMI that could lead to a magnon gap opening at the K-point, such as dipolar, Kitaev and magnon-phonon interactions:

(i) Dipolar interactions are long-ranged but much weaker than the magnetic exchange interactions, so their effect is usually seen for rather small wave vectors in the vicinity of the Γ-point. Even if they did lift the magnon degeneracy at the K-point, their intrinsic weakness could not account for the observed magnitude of the gap.

(ii) Kitaev interactions were proposed for instance in ref. 34 to explain measurements on $CrI_3$ but are ruled out for $Mn_5Ge_3$ both by our simulations and by general considerations. The interactions extracted from our DFT calculations include both the Heisenberg exchange, the DMI and the symmetric anisotropic exchange (SAE), which includes the Kitaev interaction. The SAE was found to be rather weak and unable to open the observed magnon gap at the K-point. The weakness of the SAE (and so of potential Kitaev interactions) could be anticipated from the weak magnetic anisotropy measured for this system. This reflects the lack of heavy elements in the material that could supply a strong spin-orbit interaction, which is a key ingredient for obtaining a sizeable Kitaev interaction. We are also not aware of any Kitaev material candidates containing only elements from the first four rows of the periodic table (i.e., $Z < 36$), likely due to the preceding reason. To make this argument more quantitative, we employ the theory of magnetic exchange interactions for systems with weak SOC presented by Moriya[30], Eqs. 2.3 and 2.4. The DMI is first-order in the weak SOC, while the Kitaev interaction is part of the SAE which is second-order in SOC and so is much weaker than the DMI. The Kitaev interaction, if present, would contribute to the magnetic anisotropy energy, which is about 1 meV/unit cell for $Mn_5Ge_3$ (the DMI does not contribute to the magnetic anisotropy energy of the ferromagnetic state). To give an estimate of the potential magnitude of the Kitaev interaction using only experimental input, we distribute the magnetic anisotropy energy on one of the set of bonds for which we identified the DMI, bond #2 indicated in Fig. 1b. This set of bonds occurs four times in the unit cell, as it connects each Mn1 site to its six Mn2 neighbours, and so could have 1 meV/24 = 0.04 meV maximum Kitaev strength. The SAE obtained directly from the DFT calculations is about 0.02 meV in magnitude, which is in line with this estimate, and is one order of magnitude smaller than the values found for the DMI (0.57 meV for the set of bonds #2), as expected from the theory of the magnetic exchange interactions for systems with weak SOC.

(iii) Magnon-phonon interactions can result in gaps at the crossing points between the magnon and phonon branches. However, our INS data ruled out this possibility. The measured excitations at different **Q** vectors around and at the K-point are solely of magnetic origin. This has been verified through their temperature dependence. All the peaks observed in the energy range from 10 to 20 meV at $T = 10$ K are replaced by a broad quasi-elastic signal (centered at 0 meV) above the ordering temperature as shown in Fig. 3a. Hence no phonon modes were detected in the vicinity of

the K-point with an energy compatible with that of the magnons, which is a requirement for the gap opening mechanism through magnon-phonon interactions.

Therefore, we can assert that the only reasonable mechanism for the gap opening at the K-point is the DMI.

### Simulation of magnon surface states
Here we explore the expectation that if the bulk magnon band structure has some topologically non-trivial character it should be accompanied by magnon surface states. To do so, we compute the magnon band structure of slabs which are finite in one direction and periodic (i.e., infinite) in the other two directions. Comparing the simulations performed for the same slab with periodic and open boundary conditions along the chosen surface normal enables us to identify the energy range corresponding to the surface projection of the bulk magnon bands. Surface magnons are then expected to appear in the regions of $(E, \mathbf{q})$ where bulk magnon bands are absent.

To illustrate this point, we performed simulations using the simplified spin model depicted in Fig. 4a with parameters fitted to the experimental measurements in Fig. 2b. We created a rectangular supercell and extended it by 20 unit cells along the $(01\bar{1}0)$ direction of the original hexagonal lattice. The chosen path in reciprocal space is perpendicular to the surface normal and shown in Fig. 4b. Figure 4c shows that indeed there is a pocket in the K–M−K path and with energies between 10 and 12 meV from which bulk magnons are absent, with or without DMI. Figure 4d shows the modified magnon band structure upon truncating the crystal along the $(01\bar{1}0)$ direction, i.e., making a horizontal cut through the lattice shown in Fig. 4a. We indeed find that magnon surface states do appear in the identified region where bulk magnon bands were absent. Without the DMI, these surface magnons are disconnected and gapped from each other. The DMI restructures the band connectivity and leads to a crossing that resembles a distorted Weyl-like crossing. The crossing is not located at the M-point as the slab loses the $ac$-mirror plane, retaining instead a twofold rotation around the $c$-axis. There are other surface magnons at around 3 meV and 18 meV that are only weakly affected by the DMI, and result from the reduced coordination number introduced by creating the surface. Lastly, the identified surface magnons were found to extend only a couple of unit cells towards the interior of the slab, confirming their localization at the surface.

The experimental detection of the predicted surface magnons is quite challenging, as the scattering volume is too small for a straightforward detection using INS. Other techniques such as Brillouin light scattering (for magnons near the Γ-point) or electron energy loss spectroscopy could be considered for this purpose[35].

## Discussion
We now briefly discuss some outstanding points. Firstly, we return to the quantitative disagreement between theory and experiment concerning the magnon bands. The main issue seems to be an overestimation of the magnitude of the magnetic interactions in the DFT calculations, which has also been found in other studies. A recent example from the literature is $Co_3Sn_2S_2$[36], where two different DFT approaches are compared with experiment, with disagreements also in the Γ − M − K − Γ plane but not in the Γ − A direction. Another issue is that the simplified spin model does not adequately capture the relative intensities of the two peaks around the K-point, despite providing a reasonable description of the experimental magnon energies. This is likely due to the model assumptions, namely treating the Mn2 sites as a single effective spin and neglecting the atomic magnetic form factor, as well as employing a simple Lorentzian broadening for the peaks. This shows the need for further developments on the theoretical side. We also noticed that the ratio of the intensities of the two peaks varies slightly in different experimental setups, see Figs. 3b, c. This might be

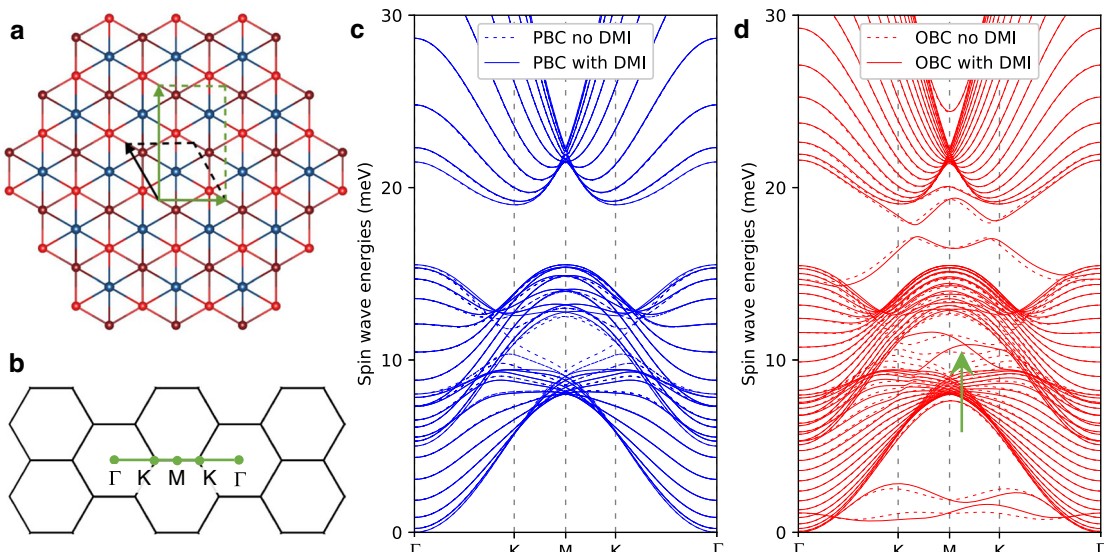

**Fig. 4 | Magnon surface states in a (01$\bar{1}$0)-oriented slab of Mn$_5$Ge$_3$. a** Relation between the primitive cell and the rectangular cell used to define the slab, viewed along the *c*-axis, using the simplified model that fits the experimental data in Fig. 2b. The Mn1 chains are indicated by the two different shades of red, while the effective spin taken to represent the Mn2 sites is shown in blue. **b** Path in reciprocal space used for the magnon calculation. **c** Magnon bands of a 20-unit-cell-wide slab computed with periodic boundary conditions (PBC). **d** Magnon bands of a 20-unit-cell-wide slab computed by changing periodic to open boundary conditions (OBC) along the (01$\bar{1}$0) orientation. Results obtained excluding/including the Dzyaloshinskii-Moriya interaction (DMI) are shown by dashed/solid lines. The green arrow in **d** indicates the DMI-enabled magnon surface state.

due to changes in the domain state of the sample arising from the measurement history.

Our INS measurements for Mn$_5$Ge$_3$ revealed a peculiar dispersion for the second mode in the Γ − K direction. Such a steep linear-like dispersion close to the Γ-point resembles an AFM magnon or a phonon mode. We restate that the material is FM and the measured excitations are of magnetic nature as verified by measurements above $T_C$ (Fig. 3a and Supplementary Figs. 4–8), so that both simple explanations are ruled out. However, it is possible to have formation of hybrid collective modes. INS can identify the magnetic and phononic component of such excitations, which are referred as magnon-polarons (magnetoelastic modes), and are reported in several magnetic materials[37,38]. Avoided crossings are the common signature of magnetoelastic interactions, but they also underpin the magnetovibrational scattering term in the INS scattering cross section[39,40]. Additional discussion is given in the Supplementary Information, Section I.C. We propose that Mn$_5$Ge$_3$ is also an interesting 3D FM candidate material for detailed investigation of magnetoelastic effects[41,42].

The key interaction responsible for opening the magnon gap at the K-point and thus endowing the Dirac magnons with a topological character is the DMI. There are several possible routes by which this interaction could be engineered, so that the magnon properties can be tuned for specific purposes for magnonic devices operating in a broad temperature range. Firstly, chemical substitution of Ge by Si has been explored in the literature to connect to the multifunctional AFM Mn$_5$Si$_3$[43–46]. Substituting Ge by Si reduces $T_C$[47], but the impact on the magnons and on the DMI is unknown. On the other hand, carbon implantation is demonstrated to enhance $T_C$[27,48] for which the imprint on the Heisenberg exchange interactions has been theoretically established[29], but once again the effect on the magnons and the DMI remains unexplored. Lastly, Mn$_5$Ge$_3$ can also be grown in thin film form[27,48]. This could modify the DMI by epitaxial strain, which would also be interesting in connection to potential magnetoelastic effects, by interfaces to other materials, or by quantum confinement effects if the thickness is just a few nanometers.

In conclusion, we presented a combined theoretical and experimental study of the magnons in the centrosymmetric 3D FM

Mn$_5$Ge$_3$. Despite the inversion symmetry, significant DMI has been theoretically identified on Mn-Mn bonds which do not contain an inversion center. This DMI is responsible for opening a gap in the magnon spectrum at the K-point, where otherwise symmetry would enforce a Dirac-like crossing of the magnon bands. INS measurements of the magnon spectrum show qualitative agreement with the main points predicted by theory, and confirm the expected gap at the K-point. We experimentally observe the closing of the gap by rotating the magnetization from the *c*-axis to the $a^*$-axis with a magnetic field. This both validates the gap generation mechanism and the topological nature of the magnons at the K-point, thus establishing Mn$_5$Ge$_3$ as a realization of a gapped Dirac magnon material in three dimensions. The ability to control the gap at the K-point with an external magnetic field will also impact topological magnon surface states, and deserves further study. As the macroscopic magnetic properties of Mn$_5$Ge$_3$ can be tuned by chemical substitution of Ge with Si or by carbon implantation, and it can also be grown as thin films in spintronics heterostructures, we foresee that the features of the newly-discovered topological magnons can be engineered and subsequently integrated in novel device concepts for magnonic applications. Looking beyond Mn$_5$Ge$_3$, the physical mechanism leading to the formation of topological magnons at the K-point should be present in many other bulk centrosymmetric hexagonal materials, which opens an exciting avenue for future investigations.

## Methods

### Experimental methods
Inelastic neutron scattering (INS) experiments have been carried out on a cold (IN12) and a thermal (IN22) triple axis spectrometer at the Institut Laue-Langevin, in Grenoble, France. We use the hexagonal coordinate system and the scattering vector **Q** is given in reciprocal lattice units (r.l.u.). The wave-vector **q** is related to **Q** through **Q** = **q** + **G**, where **G** is a Brillouin zone center. Inelastic scans were performed with constant |**k**$_f$|, where **k**$_f$ is the wave-vector of the scattered neutron beam. Data were collected holding either the energy (constant-*E*) or the scattering vector (constant-**Q**) fixed. Further details on the

experimental procedures and additional measurements can be found in the Supplementary Information, Section I).

## Theoretical methods

The theoretical results were obtained with DFT calculations and the extracted spin Hamiltonian. The unit cell parameters and the atomic positions were optimized with the DFT code Quantum Espresso[49]. The magnetic parameters were computed with the DFT code juKKR[50,51], which are then used to solve a spin Hamiltonian in the linear spin wave approximation[52]. Further details on all these aspects can be found in the Supplementary Information, Section II).

## Data availability

The authors declare that the data supporting the findings of this study are available within the paper, its supplementary information file and in the following repositories[54–58].

## Code availability

The DFT simulation packages Quantum Espresso and juKKR are publicly available (see Methods). The code for the solution of the linear spin wave problem is available from the corresponding authors upon request.

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

## Acknowledgements

We thank S. Raymond for discussions and comments. The work of M.d.S.D. made use of computational support by CoSeC, the Computational Science Centre for Research Communities, through CCP9. N.B. acknowledges the support of JCNS through the Tasso Springer fellowship. F.J.d.S. acknowledges support of the European H2020 Intersect project (Grant no. 814487), and N.M. of the Swiss National Science Foundation (SNSF) through its National Centre of Competence in Research (NCCR) MARVEL. This work was also supported by the Brazilian funding agency CAPES under Project No. 13703/13-7 and the Deutsche Forschungsgemeinschaft (DFG) through SPP 2137 "Skyrmionics" (Project LO 1659/8-1). We gratefully acknowledge the computing time granted through JARA on the supercomputer JURECA[53] at Forschungszentrum Jülich GmbH and by RWTH Aachen University. The neutron data collected at the Institut Laue-Langevin are available at refs. 54–57.

## Author contributions

M.d.S.D., N.B. and F.J.d.S. contributed equally to this work. M.d.S.D. and N.B. conceived the project together with T.B. and S.L. M.d.S.D. performed most DFT calculations and the spin-wave modelling, with additional calculations performed by F.J.d.S. J.P. grew the $Mn_5Ge_3$ single crystal. N.B. performed the experimental measurements and the corresponding data analysis. K.S. and F.B. were local contacts of IN12 and IN22 and provided instrument support. The theoretical aspects of the work were discussed with N.M., S.B. and S.L. All authors participated in the discussion of the results. M.d.S.D., N.B. and F.J.d.S. wrote the manuscript with input from all authors.

## Funding

## Competing interests

The authors declare no competing interests.
