## [Peer Review File · Nature Communications]

Topological magnons driven by the Dzyaloshinskii-Moriya interaction in the centrosymmetric ferromagnet Mn₅Ge₃Reviewers' comments:

Reviewer #1 (Remarks to the Author):

The manuscript authored by M. dos Santos Dias et al. presents a comprehensive investigation involving both theoretical and neutron-scattering approaches to study the topological magnetic excitations in the centrosymmetric ferromagnet Mn₅Ge₃. The authors propose the existence of a gapped Dirac magnon arising from the DMI interaction. This claim is supported by an analysis of exchange interactions through DFT calculations and the linearized semiclassical equation of motion.

The paper's most intriguing aspect is the identification of a magnon gap closure at the K-point with the application of a horizontal magnetic field. While the data presented in Figure 3b is compelling, this observation is not directly reproduced using either the original model or the simplified model proposed in the paper. This omission raises sufficient concerns and prevents me from recommending the publication of this work.

An important point regarding DFT-calculated parameters is buried in the SI, specifically "the calculated magnetic interactions from DFT are long-ranged and only the first few values are given in the table." This should be included in the caption of Table I. A figure showing exchange parameters as a function of distance should be provided in the SI to support the claim.

What is the reason for just showing the result up to J₆ in Table I? How important are the interactions beyond J₆? Have the authors explored the possibility of fitting the spectra using the original model? Not just performing spinwave calculation using DFT-calculated parameters but performing fitting like they did using the simplified model.

The simplified model also has six parameters (eq. S77). I am wondering if such "simplified" model is really necessary.

I suggest incorporating heatmap versions of Figure 2 for both the calculations and the experimental data in the SI for a direct comparison of INS intensities.

In conclusion, while the manuscript presents intriguing findings concerning topological magnetic excitations in Mn₅Ge₃, the aforementioned concerns and suggestions should be addressed before I can fully recommend this work for publication.

Reviewer #2 (Remarks to the Author):

The manuscript reports a combined theoretical and experimental study on the magnons in a centrosymmetric ferromagnetic material Mn₅Ge₃. The authors claim that this material is possibly a gapped Dirac magnon material and the Dzyaloshinskii-Moriya interaction is responsible for the gap opening at the K-point in the magnon spectrum. They further demonstrate that the gap at the K-point can be closed by rotating the magnetization from the c-axis to the a-axis. Based on the calculation of the magnon surface states, the authors conclude that the magnon in Mn₅Ge₃ should have a topological nature.

The exploration of topological magnon materials through experiments is indeed interesting.

However, as the authors have acknowledged, many topological magnon materials have already been identified, including the gapless/gapped Dirac magnons in both collinear and noncollinear magnets. This manuscript, while introducing a new type of material, falls short of being deemed a significant breakthrough. Hence, in my opinion, the findings presented in this manuscript lack the necessary innovative aspects that warrant publication in Nature Communications.

Furthermore, the main conclusions in this manuscript are derived from the comparison of experiments and theories, rather than direct evidence from experiments. However, I find that the experimental and theoretical results do not match well, which is insufficient to support their conclusion. My judgment is based on the following points.

1) It is important to consider not only the magnon dispersion but also the distribution of spectral weights when determining the agreement between theoretical results and experimental observations. Upon comparing the experimental spectra with the theoretical ones, I have noticed inconsistencies between the theoretical results and experimental observations. Specifically, (i) along the K - Γ direction, the experimental results in Fig. 2(a) of the main text indicate that the spectral intensity of the two low-energy modes is significantly stronger than that of the third high-energy mode, while the theoretical results in Figure S14 of the Supplementary Material show the opposite trend; (ii) at the K point, Fig. 3(b) demonstrates that the spectral intensity of the low-energy mode is stronger than that of the high-energy mode, which contradicts the theoretical result shown in Fig. S14.

Therefore, I do not believe that the model they proposed is a suitable low energy effective model for describing this material. Consequently, their claim that the Dzyaloshinskii-Moriya interaction plays a crucial role in this material is also lacking solid evidence. The author should try to examine other models to see if they can provide results that are more consistent with the experiment.

2) The author has excluded other potential interactions that could contribute to the K -point energy gap through their DFT calculations. However, it is worth noting that the K -point energy gap was found to be very small, and the Dzyaloshinskii-Moriya interaction they provided was also very small. As a result, other interactions, such as the Kitaev interaction, which could potentially account for the K -point energy gap, cannot be ruled out solely based on DFT calculations due to the computational accuracy.

Furthermore, the authors argued that the absence of heavy elements with strong spin-orbit coupling suggests the lack of a sizeable Kitaev interaction in this material. However, it is important to note that the Dzyaloshinskii-Moriya interaction also relies on spin-orbit coupling. Given that the author believes that the spin-orbit coupling in this material is sufficient to generate the required Dzyaloshinskii-Moriya interactions, it raises the question of why the same spin-orbit coupling cannot provide the required Kitaev interaction.

3) I also have a question on the magnetic anisotropy energy K in the model. Why is the magnetic anisotropy energy K uniform in the model (1), even though Mn1 and Mn2 are clearly not equivalent?

Reviewer #3 (Remarks to the Author):

Dos Santos Dias and his collaborators present an insightful study involving inelastic neutron scattering and theoretical calculations of the magnon spectrum within the centrosymmetric ferromagnet Mn₅Ge₃. The authors experimentally investigate the magnon spectrum, focusing on the gap observed at the K-point. This gap is found to close upon rotating the ordered spin moment from the out-of-plane direction to the in-plane direction using an external magnetic field, suggesting its non-trivial topological nature.

The experimental results exhibit reasonable agreement with the theoretical calculations. The authors have also addressed the challenges associated with accurately computing magnetic interactions in real crystals due to their high sensitivity to structural parameters. Despite the centrosymmetric nature of Mn₅Ge₃, the authors identified the influence of a finite Dzyaloshinskii-Moriya interaction (DMI), stemming from the absence of an inversion center at the midpoint of corresponding bonds. This DMI effectively lifts the degeneracy of two magnon modes at the K-point, resulting in the observed gap. Additionally, the authors evaluate alternative mechanisms that could account for the observed gap, such as dipolar interaction, Kitave interaction, and magnon-phonon interaction. These mechanisms are excluded due to their inability to explain the observed gap size or the absence of appropriate phonon modes near the K-point. Furthermore, the authors identify magnon surface states through simulations, enhancing the depth of their findings.

The topic of topological magnons is highly interesting for the quantum magnet community. This study introduces a new gapped Dirac magnon material, the manuscript is thoughtfully presented, and the supplementary material also provides valuable information about the experimental details, data analysis, and theoretical approach. I recommend its publication in Nature Communications.

However, I would like to suggest a minor improvement: including the uncertainties for all values listed in Table 1 would be beneficial to strengthen the rigor of this work.

I commend the authors for their comprehensive exploration of topological magnon excitations in Mn₅Ge₃. The manuscript effectively combines experimental and theoretical approaches, providing valuable insights into the field. With the suggested minor revision, the paper will be well-prepared for publication.

Response to the reviewers

In the following, we address the concerns of the reviewers point by point. The original reports are shown in black, and our replies are in blue. Changes to the manuscript and the supplement are highlighted in red.

Report of Reviewer #1

The manuscript authored by M. dos Santos Dias et al. presents a comprehensive investigation involving both theoretical and neutron-scattering approaches to study the topological magnetic excitations in the centrosymmetric ferromagnet Mn_5Ge_3 . The authors propose the existence of a gapped Dirac magnon arising from the DMI interaction. This claim is supported by an analysis of exchange interactions through DFT calculations and the linearized semiclassical equation of motion.

The paper's most intriguing aspect is the identification of a magnon gap closure at the K-point with the application of a horizontal magnetic field. While the data presented in Figure 3b is compelling, this observation is not directly reproduced using either the original model or the simplified model proposed in the paper. This omission raises sufficient concerns and prevents me from recommending the publication of this work.

This is a very important remark that substantiates the reviewer's decision not to support our manuscript for publication, and its incorrect conclusion results from a misunderstanding. We quote from our manuscript: "However, it is known that the impact of the DMI on the FM magnon spectrum depends on the relative alignment between the vectors that characterize this interaction and the FM magnetization [32–34]. Adapting these arguments to the hexagonal symmetry of Mn_5Ge_3 leads to the prediction that the magnitude of the gap should depend on the relative alignment of the FM magnetization and the crystalline c -axis, and in particular should vanish if the two are perpendicular." This is a feature of both the DFT-based and the simplified models, as it results from the symmetry of the crystal which is imparted on the orientations of the DMI vectors. The magnon dispersions computed with DMI and the magnetization along the a -axis are identical to the ones shown in Fig. 1c and Fig. 2b, which were obtained by excluding the DMI from the calculations. We have added a sentence to the first paragraph of the subsection "Closing the topological gap" to highlight this fact. As explained in the text, the most important aspect of the applied field is that it can rotate the magnetization, as its direct impact on the magnon energies is only a very small Zeeman shift, which is confirmed from the measurements with the applied field along the c -axis.

An important point regarding DFT-calculated parameters is buried in the SI, specifically "the calculated magnetic interactions from DFT are long-ranged and only the first few values are given in the table." This should be included in the caption of Table I. A figure showing exchange parameters as a function of distance should be provided in the SI to support the claim.

We agree with the reviewer that this is an important point and deserves further elaboration. Following their advice, we have expanded the table caption as recommended, and we added a new figure, S11, to the SI reporting the exchange interactions as a function of distance.

What is the reason for just showing the result up to J_6 in Table I? How important are the interactions beyond J_6 ? Have the authors explored the possibility of fitting the spectra using the original model? Not just performing spinwave calculation using DFT-calculated parameters but performing fitting like they did using the simplified model.

We showed only interactions up to J_6 in order to compare with the calculation results listed in Ref. 29 and for the sake of conciseness. Within the DFT parameter set, the interactions beyond J_6 are very important, as seen in the new figure S11 in the supplementary material, and have been included in all our calculations of the magnon dispersions. We then introduced the simplified model to try to capture the complexity of the experimentally determined magnon dispersions in a much simpler picture. We also considered attempting a direct fit of the experimental data to the full model with ten instead of only three Mn sites in the unit cell but found that the increased complexity was not easily tractable with a robust fitting approach. When posed in this way the problem seems to be underdetermined, as we do not have experimental access to the higher-energy ($E > 40$ meV) magnon modes.

The simplified model also has six parameters (eq. S77). I am wondering if such “simplified” model is really necessary.

As we explained in the previous reply, the simplified model was introduced to provide a tractable means of directly fitting the experimental data. Besides following the overall experimental dispersion quite well, it’s important that it also reproduces the observed magnon gap at the K-point using a DMI of comparable magnitude to the one obtained with DFT. One virtue of the simplified model is that it is only fitted to the magnon energies at the high-symmetry points. The magnon dispersions between the high-symmetry points can then inform on how good or bad the assumptions used to define the simplified model actually are. We interpret the overall good agreement with the measurements as evidence that approximating the Mn2 sites as a single effective spin does not compromise too much the description of the low-energy magnon modes, making it a meaningful simplification.

I suggest incorporating heatmap versions of Figure 2 for both the calculations and the experimental data in the SI for a direct comparison of INS intensities.

We have previously attempted to make such a comparison but could not arrive at a very insightful picture. From the experimental side, as the data was collected with either a cold or a thermal triple-axis spectrometer and not with a single time-of-flight instrument, the actual amount of data available to build such a heatmap is too sparse; we did attempt this and little can be discerned from the interpolated heatmap. From the theoretical side, there is lack of information about the broadening of the magnon bands which also affects the computed INS intensities, while the assumptions used to build the simplified model, namely treating the Mn2 sites as a single effective spin, will also result in discrepancies. To summarise, we could not establish a reasonable procedure to post-process experimental and theoretical heatmaps that could be meaningfully compared.

In conclusion, while the manuscript presents intriguing findings concerning topological magnetic excitations in Mn_5Ge_3 , the aforementioned concerns and suggestions should be addressed before I can fully recommend this work for publication.

We thank the reviewer for reading our manuscript and for the questions and remarks which led to several improvements in our presentation. We believe that we have convincingly addressed all the concerns and suggestions, in particular the putative omission of key simulations concerning the closing of the magnon gap at the K-point, and we hope that they will now support the publication of our work.

Report of Reviewer #2

The manuscript reports a combined theoretical and experimental study on the magnons in a centrosymmetric ferromagnetic material Mn_5Ge_3 . The authors claim that this material is possibly a gapped Dirac magnon material and the Dzyaloshinskii-Moriya interaction is responsible for the gap opening at the K-point in the magnon spectrum. They further demonstrate that the gap at the K-point can be closed by rotating the magnetization from the c -axis to the a -axis. Based on the calculation of the magnon surface states, the authors conclude that the magnon in Mn_5Ge_3 should have a topological nature.

The exploration of topological magnon materials through experiments is indeed interesting. However, as the authors have acknowledged, many topological magnon materials have already been identified, including the gapless/gapped Dirac magnons in both collinear and noncollinear magnets. This manuscript, while introducing a new type of material, falls short of being deemed a significant breakthrough. Hence, in my opinion, the findings presented in this manuscript lack the necessary innovative aspects that warrant publication in Nature Communications.

We agree with the reviewer that several publications have by now arisen in the field of topological magnons. However, the unique aspect of our work is demonstrating the closing of the topological magnon gap for a 3D ferromagnet, not for a 2D one like CrI_3 , which is relevant both for fundamental and practical reasons. Topological properties (of magnons or other quasiparticles) strongly depend on the dimensionality, so having a 3D material system in which these can be explored is very valuable from a fundamental point of view. The ease of modifying the topological properties in a 3D material, namely controlling the magnon gap by simply rotating the magnetization, is an advantage of the hexagonal symmetry of Mn_5Ge_3 , in contrast to the 3D material $\text{Lu}_2\text{V}_2\text{O}_7$, which is cubic, for which this gap closing effect has never been demonstrated. On the practical side, Mn_5Ge_3 is very stable, orders near room temperature and is easy to grow in either bulk or thin-film forms, overcoming the limitations of 2D magnets such as CrI_3 which can be hard to assemble and degrade under atmospheric conditions. Lastly, as we concluded in the manuscript: “Looking beyond Mn_5Ge_3 , the physical mechanism leading to the formation of topological magnons at the K-point should be present in many other bulk centrosymmetric hexagonal materials, which opens an exciting avenue for future investigations.” We believe that the blueprint for testing the topological character of a magnon gap which we presented in our work will indeed trigger a reassessment of many such materials.

Furthermore, the main conclusions in this manuscript are derived from the comparison of experiments and theories, rather than direct evidence from experiments. However, I find that the experimental and theoretical results do not match well, which is insufficient to support their conclusion. My judgment is based on the following points.

We would like to preempt this judgment. While the quantitative agreement between theory and experiment is unsatisfactory, what really matters is that the physical picture emerging from the simulations convincingly explains the experimental measurements. The closing of the magnon gap at the K-point is direct experimental evidence for the proposed physics. In fact, the experiment cannot be interpreted without the corresponding theoretical background, and this is indeed always the case.

1) It is important to consider not only the magnon dispersion but also the distribution of spectral weights when determining the agreement between theoretical results and experimental observations. Upon comparing the experimental spectra with the theoretical ones, I have noticed inconsistencies between the theoretical results and experimental observations. Specifically, (i) along the K- Γ direction, the experimental results in Fig. 2(a) of the main text indicate

that the spectral intensity of the two low-energy modes is significantly stronger than that of the third high-energy mode, while the theoretical results in Figure S14 of the Supplementary Material show the opposite trend; (ii) at the K point, Fig. 3(b) demonstrates that the spectral intensity of the low-energy mode is stronger than that of the high-energy mode, which contradicts the theoretical result shown in Fig. S14.

This is keenly observed and a very good point. We believe that this indicates limitations arising from the assumptions used to construct the simplified model, namely treating the Mn2 sites as a single effective spin and not taking into account the atomic magnetic form factor, all of which will affect the overall intensities of each magnon branch. Another source of discrepancies is lack of information about the broadening of the magnon bands which also affects the computed INS intensities. We employed the simple Lorentzian broadening given by Eq. S47 in the Supplementary Material, but this can be a poor approximation for the broadening of the peaks resolved in experiment, for instance in Fig. 2a of the main text. An explicit calculation of the magnon lifetimes would improve the simple broadening model, as well as taking into account the resolution function of the spectrometer in some fashion, but we are not in a position to account for these at present. Concerning the intensities experimentally measured at the K-point, we note that there seems to be a sort of training effect. In Fig. 3b and in zero field, the magnon mode with lower energy has a slightly higher intensity than the magnon mode with higher energy, while the zero-field data in Fig. 3c shows both modes with roughly the same intensity (the absolute count values cannot be directly compared as the sample environment is not the same for both sets of measurements). We speculate that this might be due to changes in the domain state of the sample arising from the measurement history, but reassuringly the corresponding magnon energies are unchanged.

Therefore, I do not believe that the model they proposed is a suitable low energy effective model for describing this material. Consequently, their claim that the Dzyaloshinskii-Moriya interaction plays a crucial role in this material is also lacking solid evidence. The author should try to examine other models to see if they can provide results that are more consistent with the experiment.

We concede that the simplified model might miss some aspects of the spin dynamics which are needed to fully explain the observed experimental INS intensities. However, the simplified model provides a tractable means of directly fitting the experimental data, follows the overall experimental dispersion quite well despite being fitted only to the experimental data at the high-symmetry points, and importantly it reproduces the observed magnon gap at the K-point using a DMI of comparable magnitude to the one obtained with DFT. In both the simplified and in the DFT models the sole mechanism for opening the magnon gap at the K-point is the Dzyaloshinskii-Moriya interaction. As addressed in the following replies, we have systematically ruled out other interactions as the potential origin of this magnon gap. Without concrete suggestions from the reviewer, we do not see what other models or gap opening mechanisms we might attempt to explore.

2) The author has excluded other potential interactions that could contribute to the K-point energy gap through their DFT calculations. However, it is worth noting that the K-point energy gap was found to be very small, and the Dzyaloshinskii-Moriya interaction they provided was also very small. As a result, other interactions, such as the Kitaev interaction, which could potentially account for the K-point energy gap, cannot be ruled out solely based on DFT calculations due to the computational accuracy.

We have verified that the calculations of the magnetic exchange interactions are sufficiently precise to make the reported values meaningful. It is also worth noting that experiments report

the typical magnitude of the Dzyaloshinskii-Moriya interaction to be of the order of 1 meV. For example, in Ref. 10 the value of 1.5 meV is reported for $\text{Lu}_2\text{V}_2\text{O}_7$, while Ref. 16 reports 0.31 meV for CrI_3 . While we agree that the absolute values computed from DFT can have some discrepancy in comparison with the experimentally determined values (when those are available), what is actually relevant is the relative importance of the various contributions to the magnetic exchange interactions, which we address in the next reply.

Furthermore, the authors argued that the absence of heavy elements with strong spin-orbit coupling suggests the lack of a sizeable Kitaev interaction in this material. However, it is important to note that the Dzyaloshinskii-Moriya interaction also relies on spin-orbit coupling. Given that the author believes that the spin-orbit coupling in this material is sufficient to generate the required Dzyaloshinskii-Moriya interactions, it raises the question of why the same spin-orbit coupling cannot provide the required Kitaev interaction.

This is answered by the theory of magnetic exchange interactions for systems with weak spin-orbit coupling, e.g., Moriya, Phys. Rev. 120, 91–98 (1960), Eqs. 2.3 and 2.4 [our Ref. 30]. The DMI is first-order in the weak SOC, while the Kitaev interaction is part of the symmetric anisotropic exchange which is second-order in SOC and so is much weaker than the DMI. This is verified by our explicit DFT calculations of the Kitaev interactions and other contributions to the symmetric anisotropic exchange (i.e., non-DMI). The Kitaev interaction, if present, would contribute to the magnetic anisotropy energy, which is about 1 meV/f.u. for Mn_5Ge_3 (the DMI does not contribute to the magnetic anisotropy energy of the ferromagnetic state). To give an estimate of the potential magnitude of the Kitaev interaction using only experimental input, we distribute the magnetic anisotropy energy on one of the set of bonds for which we identified the DMI, bond #2 indicated in Fig. 1b. This set of bonds occurs four times in the unit cell, as it connects each Mn1 site to its six Mn2 neighbours, and so could have $1 \text{ meV}/24 = 0.04 \text{ meV}$ maximum Kitaev strength. The typical values for the symmetric anisotropic exchange obtained directly from the DFT calculations are about 0.02 meV in magnitude, which is in line with this estimate, and are one order of magnitude smaller than the values found for the DMI (0.57 meV for the set of bonds #2), as expected from the theory of the magnetic exchange interactions for systems with weak spin-orbit coupling. We have added this argument to our discussion of the Kitaev interaction in the manuscript.

3) I also have a question on the magnetic anisotropy energy K in the model. Why is the magnetic anisotropy energy K uniform in the model (1), even though Mn1 and Mn2 are clearly not equivalent?

We have indeed computed the effective anisotropy matrices for Mn1 and Mn2 from DFT, which contain both the onsite anisotropy and the sum of all symmetric anisotropic exchange interactions, including Kitaev terms. Mn1 is in the Wyckoff position $4d$ which has 3_2 symmetry, so its effective anisotropy is of the uniaxial type. Writing $\mathcal{H} = K_z S_z^2$, our DFT calculations give $K_z \approx 0.07 \text{ meV}$ for the Mn1 sites. Mn2 is in the Wyckoff position $6g$ which has $m2m$ symmetry and is described by a biaxial anisotropy. Defining a local set of axes so that z is the c -axis and y is the normal to the mirror plane passing by the atom, we write $\mathcal{H} = K_{xy} (S_x^2 - S_y^2) + K_z S_z^2$ and our calculations resulted in $K_{xy} \approx -0.12 \text{ meV}$ and $K_z \approx -0.18 \text{ meV}$, respectively. Adding up all the contributions for a ferromagnetic state, the K_{xy} contributions from the different Mn2 sites cancel out, and we have $K_z^{\text{total}} = 4 \times 0.07 + 6 \times (-0.18) = -0.80 \text{ meV}$, which rounds up to -0.1 meV per Mn atom, as quoted in the main text. We have added this information to the supplemental material. However, we verified that the computed magnon dispersions are insensitive to these details, and that essentially the same results are obtained by assigning the same uniaxial anisotropy to all Mn sites, which we then adopted for simplicity of presentation.

Report of Reviewer #3

Dos Santos Dias and his collaborators present an insightful study involving inelastic neutron scattering and theoretical calculations of the magnon spectrum within the centrosymmetric ferromagnet Mn_5Ge_3 . The authors experimentally investigate the magnon spectrum, focusing on the gap observed at the K-point. This gap is found to close upon rotating the ordered spin moment from the out-of-plane direction to the in-plane direction using an external magnetic field, suggesting its non-trivial topological nature.

The experimental results exhibit reasonable agreement with the theoretical calculations. The authors have also addressed the challenges associated with accurately computing magnetic interactions in real crystals due to their high sensitivity to structural parameters. Despite the centrosymmetric nature of Mn_5Ge_3 , the authors identified the influence of a finite Dzyaloshinskii-Moriya interaction (DMI), stemming from the absence of an inversion center at the midpoint of corresponding bonds. This DMI effectively lifts the degeneracy of two magnon modes at the K-point, resulting in the observed gap. Additionally, the authors evaluate alternative mechanisms that could account for the observed gap, such as dipolar interaction, Kitaev interaction, and magnon-phonon interaction. These mechanisms are excluded due to their inability to explain the observed gap size or the absence of appropriate phonon modes near the K-point. Furthermore, the authors identify magnon surface states through simulations, enhancing the depth of their findings.

The topic of topological magnons is highly interesting for the quantum magnet community. This study introduces a new gapped Dirac magnon material, the manuscript is thoughtfully presented, and the supplementary material also provides valuable information about the experimental details, data analysis, and theoretical approach. I recommend its publication in Nature Communications.

We thank the reviewer for this accurate summary of our work and for recognising its relevance and significance.

However, I would like to suggest a minor improvement: including the uncertainties for all values listed in Table 1 would be beneficial to strengthen the rigor of this work.

The values listed in Table 1 are significant to the displayed decimal precision. We added this remark to the caption.

I commend the authors for their comprehensive exploration of topological magnon excitations in Mn_5Ge_3 . The manuscript effectively combines experimental and theoretical approaches, providing valuable insights into the field. With the suggested minor revision, the paper will be well-prepared for publication.

We thank the reviewer for appreciating our work and for supporting its publication.

REVIEWER COMMENTS

Reviewer #1 (Remarks to the Author):

The revised manuscript has improved its clarity. I think it contains sufficient novelty to be considered at Nature Communication. I have a few follow-up comments for the authors to consider.

1. It should be made clear in Fig 1 or its caption that the field is applied along the c-axis.
2. The following comment provided in the authors' reply should be included in the main text. "The magnon dispersions computed with DMI and the magnetization along the a-axis are identical to the ones shown in Fig. 1c and Fig. 2b, which were obtained by excluding the DMI from the calculations."
3. It would be good to produce the simulated result for the same cut in Fig. 3 to demonstrate the field-induced rotation of magnetization and consequently closing the gap.
4. The simplified model does not produce correct relative intensities. What about the full DFT model?

Reviewer #2 (Remarks to the Author):

The author's response and revised version of the manuscript have adequately addressed my concerns. I am satisfied with the changes made and recommend the publication of this work in Nature Communications.

However, the revised manuscript lacks sufficient explanations regarding the inconsistency between theoretical and experimental results in the intensity distribution of spin excitation spectra. I suggest that the author provide appropriate explanations to address this issue. The difference between the experimental and theoretical results is not merely quantitative, but qualitative as well. It is possible that different spin models can yield similar magnon dispersion, while exhibiting distinct intensity distributions in the excitation spectra. I agree with the author's explanation in the response that the inconsistency between theoretical and experimental results may arise from the analysis of experimental data rather than the theoretical model.

Reviewer #3 (Remarks to the Author):

Overall, the authors' responses to the referees' comments are satisfactory. It is an essential and trustworthy experimental result to demonstrate the closing of the topological magnon gap via a magnetic field for the 3D ferromagnet Mn₅Ge₃. The revised manuscript provides more details with improved scientific rigor. I recommend its publication.

Response to the reviewers

In the following, we address the concerns of the reviewers point by point. The original reports are shown in black, and our replies are in blue. Changes to the text are highlighted in red.

Report of Reviewer #1

The revised manuscript has improved its clarity. I think it contains sufficient novelty to be considered at Nature Communication. I have a few follow-up comments for the authors to consider.

We thank the reviewer for reappraising our work and supporting its publication.

1. It should be made clear in Fig 1 or its caption that the field is applied along the c -axis.

The simulations reported in Fig. 1 do not consider an external field, but we agree that it is important to add the following clarification to its caption: **The magnetization is set along the c -axis, which is the easy-axis of the material.**

2. The following comment provided in the authors' reply should be included in the main text. "The magnon dispersions computed with DMI and the magnetization along the a -axis are identical to the ones shown in Fig. 1c and Fig. 2b, which were obtained by excluding the DMI from the calculations."

We have now included this statement in the Results, first paragraph of the subsection "Closing the topological magnon gap". We also made a correction to the actual orientation of the in-plane magnetic field: it is along the a^* -axis and not along the a -axis (which differ in orientation by 30° due to the hexagonal crystal structure).

Fig. R1: Simulated constant-q inelastic neutron scattering for $\mathbf{Q} = (2.333, 0.333, 0)$ (a) using the simplified spin model and (b) using the magnetic interactions obtained from DFT with the experimental crystal structure.

3. It would be good to produce the simulated result for the same cut in Fig. 3 to demonstrate the field-induced rotation of magnetization and consequently closing the gap.

We have prepared a figure that shows the requested cuts, Fig. ??, and included it as Supplementary Figure S16 with a small paragraph of explanation. We opted not to attempt a direct

comparison with the experimental data, due to the previously discussed uncertainties in how to quantitatively model the broadening of the modes (which goes beyond the present scope of the theory), besides the limitations introduced by the assumptions inherent to the construction of the models themselves. However, in both cases we find that the gap between the two magnon modes indeed closes for $\mathbf{H} \parallel \mathbf{a}^*$, as predicted on general ground and also found in our experimental measurements.

4. The simplified model does not produce correct relative intensities. What about the full DFT model?

As can be seen in Fig. ??, when $\mathbf{H} \parallel \mathbf{c}$ both models predict a higher intensity for the higher-energy magnon mode, but this is much more pronounced in the simulation employing the magnetic interactions obtained from DFT calculations. This seems to suggest that the simplified model might be closer to describing the actual spin dynamics of the material as experimentally measured.

Report of Reviewer #2

The author's response and revised version of the manuscript have adequately addressed my concerns. I am satisfied with the changes made and recommend the publication of this work in Nature Communications.

We thank the reviewer for supporting the publication of our work in Nature Communications.

However, the revised manuscript lacks sufficient explanations regarding the inconsistency between theoretical and experimental results in the intensity distribution of spin excitation spectra. I suggest that the author provide appropriate explanations to address this issue. The difference between the experimental and theoretical results is not merely quantitative, but qualitative as well. It is possible that different spin models can yield similar magnon dispersion, while exhibiting distinct intensity distributions in the excitation spectra. I agree with the author's explanation in the response that the inconsistency between theoretical and experimental results may arise from the analysis of experimental data rather than the theoretical model.

We agree with the reviewer that it is worth elaborating on this point. We now summarise the matter as follows in the first paragraph of the Discussion: **Another issue is that the simplified spin model does not adequately capture the relative intensities of the two peaks around the K-point, despite providing a reasonable description of the experimental magnon energies. This is likely due to the model assumptions, namely treating the Mn2 sites as a single effective spin and neglecting the atomic magnetic form factor, as well as employing a simple Lorentzian broadening for the peaks. [...] We also noticed that the ratio of the intensities of the two peaks varies slightly in different experimental setups, see Figs. 3b-c. This might be due to changes in the domain state of the sample arising from the measurement history.**

Report of Reviewer #3

Overall, the authors' responses to the referees' comments are satisfactory. It is an essential and trustworthy experimental result to demonstrate the closing of the topological magnon gap via a magnetic field for the 3D ferromagnet Mn₅Ge₃. The revised manuscript provides more details with improved scientific rigor. I recommend its publication.

We thank the reviewer for their continued support and appreciation for our work.

REVIEWERS' COMMENTS

Reviewer #1 (Remarks to the Author):

I am satisfied with the revision and recommend its publication in Nature Communications.

Reviewer #2 (Remarks to the Author):

The authors have effectively addressed the comment I raised in the previous round of review. I believe the revised manuscript is now suitable for publication.